# Shopping festival atmospherics of China's singles day shopping festival and participants' perception: Scale development and validation

Jingyu Li[1]*, Xiaoming Wang[1], Qiuyang Gu[2]

1 School of Economics and Management, Yiwu Industrial & Commercial College, Yiwu, China, 2 School of Management, Zhejiang University of Technology, Hangzhou, China

* lijingyu6860@zjgsu.edu.cn

## Abstract

China's Singles Day Shopping Festivals (SDSFs) have continuously attracted consumers in recent years. As large-scale online sales events, their atmospherics have not yet been systematically analyzed. The present study aims to develop a measurement scale named "shopping festival atmospherics" (SFA) to capture the major dimensions of the atmospherics created by SDSFs. Following a rigorous procedure, both qualitative and quantitative approaches are employed to develop the measurement scale. Finally, three dimensions (cultural cues, social cues, and online cues) that contain eighteen attributes are extracted. The present study also identifies several novel shopping atmospheric cues that have not been mentioned in the literature. The happily crowded environment ("*Renao*" in the Chinese language) is a positive atmospheric cue for Chinese participants. A fair shopping environment is also a positive atmospherics cue as participants worry about price scams. Finally, the limitations and suggestions for future research are presented at the end of the paper.

## 1. Introduction

Singles Day Shopping Festivals (SDSFs), some of the most massive online promotional events in China, have continuously attracted Chinese consumers' participation for the past several years [1]. The date, November 11, is chosen as Chinese Singles Day because the four number "1s" convey the idea that unmarried young people feel lonely. However, it was not a traditional, official, or even popular holiday before Alibaba, one of China's e-retail giants, pioneered the 11.11 global shopping festival in 2009 [2]. On November 11, 2018, Alibaba again generated the highest one-day retail sales volume (GMV) of RMB 213.5 billion (US $30.8 billion) settled within a 24-hour period (Alibaba, 2018) [3], which was 26% growth from 2017 (Alibaba, 2017) [4]. This record was again exceeded in 2019, reaching RMB 268.4 billion (US $38.4 billion), another 26% jump from 2018's numbers [5]. In fact, the SDSFs today is not an exclusively promotional event for Alibaba. JD.com, Alibaba's major e-commerce competitor

**Data availability statement:** All relevant data are within the manuscript and its Supporting information files.

**Funding:** The author(s) received no specific funding for this work.

**Competing interests:** Enter: The authors have declared that no competing interests exist.

(JD.com, 2018) [6], and Netease.com, a new e-commerce player, have been actively involved for several years (NetEase, 2018) [7], making SDSFs China's super-large national-level shopping festivals in the real sense.

How have SDSFs attracted so many merchants and consumers for years? Among the few existing studies, we find some explanations. Wu et al. (2016) conducted in-depth interviews with Alibaba senior managers from various divisions and identified five successful strategies: cost leadership, incentivized gamification and social networks, quick and reliable payment, an intelligent logistic network, and a synergistic IT platform [1], whereas Xu et al. (2017) attributed the success of SDSFs to the herd effect [8]. Improving the variety of promotion categories and the interest of promotion activities are very effective promotion strategies during online shopping festival [9]. Information quality, product quality and cost savings can significantly improve consumer satisfaction with online shopping festivals [10]. Later, researchers found that the Double 11 online shopping festival was not only a promotional activity, but also an entertainment for participants. Collaboration between Chinese e-commerce platforms and merchants in terms of product promotion and atmospheric marketing strategies is key to the success of the SDSFs [9]. Xu, Wang, and Zhao (2020) posit that SDSFs creates unique shopping scenarios that can more effectively stimulate consumers' emotions and influence their shopping behaviors [11]. Hedonistic shopping has become the main motivation for consumers to participate in online shopping festivals, in addition to saving money [12].

Online promotional events (e.g., Click Frenzy in Australia, Cyber Monday and Prime Day in the U.S.) have become a global phenomenon since the emergence of Cyber Monday in 2005. China's e-business companies like to name their promotional events "Jie" (festival or cardinal in Chinese). In the modern Chinese language, "Jie" does not necessarily need a cultural or authoritative origin. Several extant studies confirm the positive connection between the festival setting and consumers' hedonic shopping [1,8,13–16] and impulsive shopping behaviors [12,16,17]. Practically, shopping festival promoters hope to enhance participants' happy and pleasant experiences and subsequently increase sales and profits with the help of festival settings. In fact, consumers do not always participate in shopping events because of hedonic motivations. Most of the time, deep discounts and monetary incentives are the major motivations. Lennon, Johnson, and Lee (2011) [18] and Swilley and Goldsmith (2013) [15]reported that participants even misbehave on Black Monday when competing for limited-time deeply discounted items.

Kolter (1973) initiated the literature stream on retailing environments [19]. The concept of "atmospherics" is used to describe the intended atmosphere settings retailers create to attract consumers. In addition to the ordinary physical atmospherics (e.g., layout, music, and color), festival atmospherics include cultural cues, such as Christmas music and scents for the Christmas shopping season [20,21] and festival food products for the Chinese New Year Festival shopping season [16]. The atmospherics of online promotional events include convenience and simplicity for Cyber Monday [15], in addition to the ordinary web atmospherics of informativeness, effectiveness and entertainment [22]. The influence of shopping festival atmospheres on consumer

behavior in SDSFs has also been the subject of research by scholars in the field. In a study conducted by Yang, Li, and Zhang (2018), it was discovered that the atmosphere surrounding SDSFs was negatively correlated with consumers' intentions to engage in sustainable consumption [23]. This atmosphere has become SDSFs most important marketing tool for stimulating consumer purchases [24]. The shopping festival atmosphere plays a positive moderating role between promotional stimulation and satisfaction, as well as between satisfaction and the intention to continue participating [25]. Despite the considerable attention that research on the atmosphere of online shopping festivals has attracted, there is still no consensus on the exact connotations of this phenomenon. There are no clear conclusions about the difference between the atmosphere of online shopping festivals and that of traditional shopping environments, and there is a lack of scientifically valid scales to describe this unique atmosphere.

"SDSFs" created by Alibaba is the earliest online shopping festival in China (2009.11.11), with the highest popularity, the most extensive consumer participation and the most influential [8,10]. The success of other online shopping festivals (e.g., JD 6.18) also draws lessons from the successful experience of SDSFs. The rapidly growing sales data of SDSFs over the years and the popular discussion on social platform [1,23]. SDSFs has undergone a significant transformation, evolving from a focus on simple price promotions to the development of comprehensive marketing campaigns that prioritize the creation of an entertaining and interactive atmosphere [26]. Given the relatively standardized and homogeneous nature of e-commerce activities worldwide, the findings of the survey on Chinese e-commerce platforms can be extrapolated to a number of other platforms globally. This offers valuable insights and recommendations for other countries preparing for similar shopping festivals. Therefore, the SDSFs is not only a typical representative of online shopping festival, but also provides reference value for the development of similar online shopping festivals.

This study posit that atmospherics represents one of the most salient characteristics of large-scale online sales events. In accordance with the nomenclature of the SDSFs, the present study endeavors to encompass the comprehensive dimensions of "shopping festival atmospherics" (SFA). Additionally, as Kolter (1973) noted that the intended atmosphere differs from the perceived atmosphere, the authors also investigate participants' perception of the atmospherics [19]. This study is expected to contribute to the literature in two ways: firstly, according to the S-O-R theory, explore the shopping festival atmosphere related to online mass sales activities, and secondly, develop a SFA (including cultural cues, social cues, and online cues) measurement scale for future research on online festival sales activities. The rest of the paper is organized as follows. In Sec. 2, we will conduct an intensive literature review on shopping atmospherics and festival shopping atmospherics. Then, the measurement scales of SFA will be developed and validated in Sec. 3. Finally, discussion and conclusions are presented in Sec. 4, followed by limitations and future research ideas in Sec. 5.

## 2. Literature review

### 2.1 Theoretical foundation

The S-O-R (Stimulus-Organism-Response) theory is derived from the S-R (Stimulus-Response) theory of behavioral psychology [27]. The S-O-R model was first proposed by Woodworth, and he explained that stimuli elicit different responses depending on the state of the organism [28]. Environmental psychologists use the S-O-R paradigm to describe the external stimulus, intervention variables, and behavioral outcomes [29]. The salient feature of the S-O-R theory is that consumer decision making goes through three distinct stages, i.e., external stimuli (S), affecting individual emotional states (O), and finally triggering behavioral responses (R) [30]. S-O-R theory is widely used to study offline and online consumer psychology and behavior. Currently, popular online consumption scenarios include: Fashion e-commerce retail [31], TikTok store [32], Live streaming e-commerce [33].

Kotler (1973) defined atmosphere as an environment designed to elicit emotional and behavioral responses from customers [19]. In 1974, Mehrabian and Russell pioneered the use of the S-O-R paradigm to examine the influence of atmospheric factors on consumer behavior [29]. In the positivist paradigm (S-O-R), numerous scholars argue that atmosphere is typically external stimulus (S) that influences the consumers' internal organism (O), subsequently leads

to behavioral response (R) [34–36]. In recent years, the S-O-R framework has been employed extensively to examine the online shopping atmosphere in a multitude of contexts. For example, the visual complexity of shopping websites [37] and the website informativeness and effectiveness of information content [38] have been identified as key factors. Chen and Li (2019) posited that the atmosphere of shopping festivities is the primary factor influencing consumer purchasing intentions [24]. Xie et al. (2023) investigated the influence of the atmosphere of an online shopping festival as a stimulus factor on consumers' intention to engage in continued participation [26]. In accordance with the S-O-R theory, Cheng and Chen (2023) devised a creative atmosphere scale, and its process and methodology provide a very valuable reference for this research [39]. This study puts forth the concept of shopping festival atmosphere (SFA) from the vantage point of consumption perception stimulus, with a foundation in the S-O-R framework. SFA is evidenced by the implementation of extensive product promotions, the configuration of online platforms, the incorporation of entertainment games, the facilitation of social interaction, and other forms of expression. SFA is a stimulating element (S), consumers experience the joy, stimulation, and excitement associated with shopping carnivals (O), ultimately exhibiting a tendency to engage in continuous participation (R).

## 2.2 Definition of atmospherics

Atmospherics refers to the conscious designing of retailing environments to arouse buyers' specific emotional responses and enhance their purchase possibility [19]. In other words, the term "atmospherics" is defined as the intentional control and structuring of environmental cues [40]. Since the early 1960s, researchers have investigated the effects of environmental elements (e.g., music and shelf space) on purchasing behaviors [40]. Kolter (1973) argued that the environmental cues rather than the products themselves play an important role in consumers' decision making because of competition and only small differences among products [19]. Almost at the same time, environmental psychologists started to use the Stimulus-Organism-Response (S-O-R) paradigm to describe environments, intervening variables, and behavioral outputs [29]. In the context of retailing, consumers' emotional response (O) plays a mediating role in the relationship between atmospheric stimuli (S) and behaviors (R). As consumers are influenced by the physical and network surroundings perceived at the point of purchase, the practice of establishing the intended atmosphere should be an effective marketing tool for retailers. How many dimensions do atmospherics contain? How are atmospherics supposed to take effect? And how do consumers perceive and respond to specific atmospherics? To answer these questions, atmospherics research has experienced several improvements during the past forty years. We classify the atmospherics literature into four categories: 1) atmospherics of physical stores; 2) atmospherics of online stores; 3) cultural atmospherics; and 4) festival atmospherics.

## 2.3 Atmospherics of physical stores

In his seminal work, Kotler (1973) posited that the store atmosphere encompasses four dimensions: visual, auditory, tactile, and olfactory [19]. In the early atmosphere literature, atmospherics almost always specifically referred to "store atmospherics". Baker (1994) classified in-store elements into ambient (e.g., music, lighting, smell), design (e.g., color, wall covering, cleanliness, layout), and social factors (i.e., salespeople) [41]. Their study posited that environmental cues are the antecedent of merchandise quality and service quality, while the latter two factors are linked to store image. Referring to an earlier review of atmospheric effects on shopping behavior [40], the store atmospheric variables were synthesized into five categories: external variables (14 items), general interior variables (14 items), layout and design variables (14 items), point-of-purchase and decoration variables (10 items), and human variables (5 items). Similarly, the store atmosphere can be divided into three dimensions: social categories (3 items), physical categories (3 items), and ambient features (2 items) [42].

In earlier research, environmental psychologists applied pleasure-arousal-dominance (PAD) to predict consumers' approach-avoidance behaviors in store atmospherics [43]. Chebat, Chebat, and Vaillant (2001) found that background music fit rather than music pleasure or music tempo-induced arousal helps salespersons attract and persuade shoppers

[44]. Later, researchers revealed that retail atmospherics can be useful in customer relationship management by creating both utilitarian and hedonic shopping values for customers [45]. In fact, store ambiance is a conscious effort to create a retail environment that makes retail more attractive to consumers than the product itself [46,47]. As retailers look to differentiate their service offerings, the atmospherics of music and aroma can be deliberately adapted to cater to young fashion shoppers' preferences [48]. Their experimental study showed that the volume of music and the presence of a vanilla aroma both significantly impact on shoppers' pleasure levels and satisfaction levels, which subsequently lead to increased time and money spent in the store.

In recent years, the field of market research has recognized color, lighting and music as atmospheric tools that can elicit better multi-sensory congruent cues (such as visual, auditory and olfactory) and ultimately influence purchase intentions [49,50]. There have been many interesting findings on the study of store atmosphere on consumer behavior. Burns, Dato-on, and Manolis (2015) reported that Hispanic consumers in the U.S. do not like crowded stores [51]. Haj-Salem, Chebat, Michon, and Oliveira (2016) investigated more than 900 North American shoppers and found that female shoppers' loyalty to malls is more affected by shopping atmospherics than that of male shoppers [52]. In young shoppers' sections and pop-up stores, playing modern and stylish music can increase their desire to stay in the mall [50]. In global apparel retailer's stores, customers in a non-English speaking country are more likely to buy when music is played in English [53]. The olfactory congruence of a shopping mall has a positive effect on bridging the mall atmosphere and brand image [54]. In young shoppers' sections and pop-up stores, playing modern and stylish music can increase their desire to stay in the mall [31]. The visual atmosphere of a coffee shop can have an impact on consumers' expectations of different coffee flavors [55].

## 2.4 Atmospherics of online stores

In the first years of the 21st century, with the emergence and booming of electronic commerce, online retailing became a new retailing format. Eroglu, Machleit, and Davis (2001) took the first step to investigate the atmospheric nature of online retailing [56]. As virtual shopping outlets, online stores in that era had no opportunity to apply VR/AR (Virtual Reality/Augmented Reality) technology, so that the online environment lacked some of the sensory appeal of traditional retail atmospherics. The entire online store environment is reduced to a computer screen. Eroglu et al. (2001) suggested that online atmospheric cues should be classified into two groups: high task-relevant cues and low task-relevant cues [56]. High task-relevant cues include descriptions, prices, terms of sale, delivery and return policies, and navigation aids (e.g., site map, guide bar), which are defined as the dimensions of site informativeness and effectiveness of information content. Low task-relevant cues include colors, borders, layout, background patterns, type styles and fonts, animation, music and sounds, which are defined as the site entertainment dimension. These atmospheric cues can create a mood or an image for the site.

Following the work of Eroglu et al. (2001) [56], Dailey (2004) proposed that navigational cues are another important dimension of web atmospherics, as restrictive navigational cues hinder consumers from using a web site, which, in turn, arouses psychological reactance and leads to avoidance behavior [57]. Similarly, Mummalaneni (2005) argued that navigation design is an important atmospheric cue for a virtual store to help consumers proceed smoothly through the purchasing stages [58]. Similar to brick-and-mortar stores, web-based stores have the atmospheric dimension of storefront designs. The online shoppers react more positively to web-based stores with a thematic and picture-based design than those with a non-thematic and text-based design [59]. A picture-based information display provides consumers a convenient store image and leads to their expectations of merchandise quality. Some scholars differentiate between the various elements that comprise the atmosphere of online stores, including information, navigation quality, as well as platform design [60,61].

With the development of web technology, multimedia controls are embedded in the online marketplace. Wu (2008) revealed that both background music and color factors have a significant effect on consumers' emotional response, which

subsequently affects their intention to purchase in online retailing settings [62]. Specifically, fast-tempo background music is linked to online shoppers' arousal emotions and pleasure more than slow-tempo music. A warm background color is linked to higher levels of arousal, whereas a cold background color is linked to higher levels of pleasure. Gao (2014) investigated the impact of website atmospheric cues (informativeness, effectiveness, and entertainment) on consumers' purchase intention and satisfaction [63]. Their empirical study indicated that flow fully mediated the atmospheric cues with regard to purchase intention and satisfaction. Khan, Wang, Ehsan, Nurunnabi, and Hashmi (2019) classified web atmospheric cues into five categories: visual cues, information cues, ethics cues, social cues, and security cues [64]. Ettis (2017) found that the blue-hues background of an online store induces more consumers' flow (Enjoyment and concentration) than yellow [65]. The unique shopping atmosphere of e-commerce live streaming is reflected in three aspects: bullet screen, guide information as well as parasocial interaction [66]. In live streaming e-commerce, participants generate high arousal when they are in an atmosphere with complex background images and fast-paced background music, and fast-paced music seems to better meet the needs of consumers [33].

## 2.5 Cultural atmospherics

Cultural factors influencing shopping atmospherics have two aspects: 1) shopping atmospherics are adapted to cultural atmospherics; and 2) shopping atmospherics adopt cultural elements. As Kolter (1973) noted in the very early stage of atmospherics research, people of different cultures have different ideas about colors, noisy environments, and aromas [19]. The establishment of shopping environments must make the target shoppers feel agreeable and comfortable. For instance, in the Chinese culture, red is a "happy" color. A red background color of a web-based retailing environment may result in a more pleasant feeling for Chinese consumers than a blue one [62]. According to the distinctiveness theory, each ethnic group has a unique community with their own cultural values. Ethnicity plays an important role in Hispanic-American consumers' shopping preferences and habits. Ethnic cues of store atmospherics are highly related to Hispanic consumers' patronage behavior [67]. Sometimes, culture creates a "must-have" for a specific segment. For example, the atmospherics for the Christmas shopping season must have the cultural cues of Christmas music, Christmas trees, and Santa Claus [20].

Second, cultural atmospherics are adopted to create differentiation. Ha and Jang (2010) reported that Korean-themed restaurants in the U.S. attract not only Korean but also American customers [68]. The Korean ethnic atmospherics, such as Korean-style interior design, music, mood, and layout, were verified to have moderating effects on the links between service and food quality and customer satisfaction and loyalty. A recent study by Naletelich and Paswan (2018) showed that toward two different art genres (realist vs. abstract), consumers' purchase intention has different antecedents [69]. The purchase decision making of consumers who are exposed to the retailing settings of realist art genres is significantly associated with product aesthetics and symbolism, whereas the purchase decision making of consumers who are exposed to the retailing settings of abstract art genres is significantly associated with hedonic and utilitarian motivations and openness to art.

## 2.6 Festival atmospherics

There are a few scattered studies in the literature on festival atmospherics and their effects on consumer behavior. However, we collect relevant references from three domains: festival shopping (and holiday shopping), shopping festivals, and large online promotions. Combined with the above review, we identify four categories of festival atmospheric cues:

**2.6.1 cultural cues.** For most retailers, Christmas is one of the most promising shopping seasons, rather than the traditional festival for the predominantly Christian western world. Erb (1985) traced back to 1891 the origin of the American Christmas as a secular holiday, a national festival of conspicuous consumption [21]. "Rudolf the Red-Nosed Reindeer" and "Here Comes Santa Claus" took the place of "Silent Night" and "Adeste Fidelis" and became the cultural cues of Christmas festival shopping. Erb (1985)' exploration raised a basic fact of culture, that is, culture means a long history, or

at least lasts a long time [21]. Culture-featured cues are those "must-have" symbols, images, and figures connected to a festival. A shopping environment decorated with cultural cues intends to sensitize consumers' cultural memory. Erb (1985) also noted the origin of the fixed days of the Christmas shopping season [21]. Consuming during the SDSFs is considered a ritualistic activity [35]. That is, something taking place regularly can also be regarded as a cultural cue.

Gift giving is regarded as an important cultural driving factor of Christmas shopping. Fischer and Arnold (1990) argued that women are probably more involved in Christmas shopping than men because women regard Christmas gift giving as a "symbolic exchange" or a "ritual" [70]. They will start Christmas gift shopping earlier in the calendar year, spend more time, and report greater success in gift selection. In many cultural contexts, gift shopping in the Christmas season is the "must-do" custom and habit, which must be deliberately considered and planned [17]. Thomas and Perters (2011) exploratory investigation of 38 American respondents indicated that Black Friday shopping is a collective consumption ritual for American shoppers [71]. Black Friday shoppers strategically schedule their shopping plans for the day as a ritualistic behavior. In addition, shopping festivals are promoted as tourism products. Cultural exploration is a motive driving tourists' visits to Dubai shopping festivals. Cultural events packaged as stage shows enhance tourists' on-site self-development [72]. The application of Confucian values ("keeping face" and "listening to others") has been observed to exert a discernible and substantial influence on consumer purchasing intentions during the SDSFs [24]. In summary, cultural cues are intended to make shopping festivals special days.

**2.6.2 social cues.** As a huge-scale collective event, shopping festivals generate heavy traffic during a limited period. In Thomas (2011) interviews, one informant mentioned that shopping on Black Friday is like "a great race" [71]. To win this type of race, shoppers even misbehave, such as pushing employees back, grabbing merchandise out of other shoppers' carts or hands, shouting at other customers, shoving other customers, fighting with other customers, tossing merchandise around, and overturning racks of merchandise [18]. Similarly, Swilley and Goldsmith (2013) noted the disorder and episodes of violence accompanying the 2014 British Black Friday and the concomitant insecurity and anxiety [15]. In some cultures, people dis like and avoid noisy or crowded environments [51]. However, the Chinese have a collectivist culture. Xu, Li, Peng et al. (2017) explained Chinese consumers crowding into Alibaba's online shopping carnival as herd behavior [8]. Whatever the motivation, the authors drew the conclusion that the factors of information incentives and social influence are crucial links to participants' pleasure. These findings suggest that Chinese consumers do not avoid noisy and crowded environments, instead feeling that more people lead to more pleasure. The "safely crowded" shopping environment is called "*Renao*" in Chinese, which means "happily crowded" environment.

Media are powerful tools to create a festival atmosphere. Summarizing in-depth interviews with Alibaba senior managers, Wu, Li, and Wei (2016) identified five critical strategies for SDSFs [1]. One is taking advantage of traditional media (e.g., newspaper, television) broadcast advertisements for the SDSFs. The red envelope ("*Hong Bao*", Chinese-style cash gift) game, Alibaba's SDSFs gala, and 24-hour live television updates on sales progress enhance the festival atmosphere. Social norms are another social environmental cue that consumers can perceive. Robinot, Ertz, and Durif (2017) found that Christmas shoppers who complied with the social norm of sustainable consumption intended to buy green Christmas products [73]. Consistent with Robinot, Ertz, and Durif (2017) [73] results, Yang, Li and Zhang (2018) found that in the context of China's SDSFs, participants' environmental concern is positively associated with attitudes toward sustainable consumption, which are mediated by participants' subjective norms [23]. The atmosphere of participation in the online shopping festival will affect consumers through the medium of social interaction with the media and with other people [24]. Extensive social interaction online and offline is also a major stimulus for a large number of consumers to participate in the SDSFs [9,25]. The social interaction atmosphere will stimulate the excitement of customers, thus enhancing the intention of consumers to continue to participate in the SDSFs [26].

**2.6.3 physical cues.** In the shopping mall context, Christmas background music, Christmas trees, Christmas advertisements, and a gala event opening the Christmas shopping season are the essential settings to establish a Christmas atmosphere [20]. The effects of ambient Christmas scents on shoppers' emotional responses are moderated

by Christmas background music [74]. That is, shoppers feel more favorable when they are exposed to both Christmas scents and Christmas music. Yeung and Yee(2010) reported that festival food products sold at a flower market will increase the probability of consumers' impulse purchase of flowers on the Chinese New Year Festival [16]. In order to enhance customer purchasing behavior, numerous retailers dedicate a considerable amount of time and resources to the implementation of distinctive merchandise displays at festivals, with the objective of attracting potential customers [75]. The authors explained that Chinese people will have an impulse to purchase flowers as they attend to have fun and enjoy the festival atmosphere.

**2.6.4  online cues.**  Although U.S. Cyber Monday is regarded as a huge-scale online promotion rather than an online shopping festival, the performance of this type of shopping event relies on online atmospherics. Swilley and Goldsmith (2013) revealed that the determinant of convenience is the antecedent of perceived usefulness and shopping enjoyment, which indicates the simplicity and clarity of online promotion atmospherics are highly related to consumers' positive participation intention [15]. Chinese researchers prefer "shopping festivals" to "large promotions" when studying such types of promotion events. An aesthetically pleasing online store design and intuitive navigation facilitate consumer engagement and enhance the shopping experience on Singles' Day [35]. It is consistent with the research conclusion of Shafiee and Bazargan (2018) [76] in shopping loyalty. Zeng, Cao, Chen, and Li (2019) investigated consumers' shopping behaviors during the rushed sales event in the 2016 SDSFs, and they found that consumers prefer mobile atmospherics because browsing and payment take fewer clicks [77]. The strong technical and service support provided by Alibaba makes consumers feel the "fairness atmospherics" of the China's Singles Day Shopping Festivals [25].

## 3.  Materials and methods

The present study adopts Churchill (1979) conventional procedures for developing measures [78]. Following Churchill (1979) eight-step measurement development paradigm, the present study posits that scale development can be divided into three principal stages: (1) initial item generation, (2) purification of the measurement scale, and (3) reanalysis of measures [78]. Additionally, this study draws upon the methodology and research process of Cheng and Chen (2023) in formulating the creative atmosphere scale [39], as detailed in Table 1. below.

**Table 1.  The Process of the SFA Scale Development.**

| Main Processes | Detailed Description | Implementation Results |
|---|---|---|
| 1.  Initial item Generation | 1. Literature Review | Online shopping festival, atmosphere |
| | 2. In-depth interview (n = 23) | Consumers participating in the SDSFs |
| | 3. Content analysis | Produce to 4 categories. |
| | 4. Content validity (2 experts) | Final results: 43 generated items. |
| 2.  Purifying the measurement scale | 1.  Refinement of instruments (panel of experts to obtain content validity) <br> 2.  Data collection <br> 3.  Purifying the measurement (Sample 1, n = 343), The EFA factor loadings and the test of reliability (Cronbach's α) were employed to reduce the scale. | Removed "Physical cues"; Removed and optimized 24 items that were not suitable for "SFA"; Final results: 19 items belonging to 3 dimensions: 1) cultural cues (6 items), 2) social cues (6 items), and 3) online cues (7 items). |
| 3.  Reanalysis of measures | 1.  Data collection (Sample 2, n = 343); The CFA was conducted with Sample 2, utilizing a three-dimensional model. <br> 2.  Criterion-related validity <br> 3.  Construct validity analysis | Reduce one item; The CFA results confirm the scale's strength, with 18 items: 1) cultural cues (6 items); 2) social cues (6 items); 3) online cues (6 items). |

Note: SDSFs = China's Singles Day Shopping Festivals; SFA = Shopping festival atmospherics; EFA = exploratory factor analysis; CFA = confirmatory factor analysis.

### 3.1 Initial item Generation

Following DeVellis (2021) 's recommendation, the authors apply three techniques to generate a sample of items: 1) A comprehensive literature review is conducted on shopping atmospherics, including not only festival atmospherics but also atmospherics of physical stores, online stores, and culture (see Sec. 2) [79]. A list of possible measurement items is generated. 2) The authors conduct semi-structured in-depth interviews to confirm possible items and add new items to the list. 3) Finally, the coarse list of measurement items is submitted to a panel of five experts with research expertise in electronic commerce. The five experts distinguish the coarse items and select the most applicable items to include in the draft of the questionnaire.

**Literature Review.** The present study first uses a literature search to specify the domain of the construct SFA. The authors identify four categories of festival atmospherics cues – cultural cues, social cues, physical cues, and online cues – from the intensive literature review on festival shopping (and holiday shopping), shopping festivals, and large online promotions (see Sec. 2).

**In-depth interviews.** The outline of the semi-structured interview includes at least the following fundamental questions: 1) Do you think the SDSFs can be regarded as a "*Jie*" (festival)? 2) Why do you think the SDSFs is a special day for you? 3) Can you feel the festival atmosphere intentionally created by the SDSFs around you? 4) Can you list some atmospheric cues you feel during the SDSFs, such as cultural cues, social cues, physical cues, or online cues? 5) What is your opinion on the atmosphere? 6) What other atmospheric cues can you share with us? In addition, to open respondents' minds, they are encouraged to share their shopping stories and experiences during SDSFs. The present study uses the technique of convenience sampling to recruit 23 respondents. Because millions of consumers participate in the SDSFs every year, it is not difficult to find the respondents. However, no additional cues emerge after the first 20 respondents. The interviews are recorded with a voice recorder app embedded in a smartphone after obtaining the respondents' consent. The voice records are transcribed into text by "*Xunfei Tingjian*", an automatic transcribing engine for the Chinese language based on cloud service.

**Content analysis.** Two research assistants corrected errors and then checked the machine transcription. Two experts coded the transcriptions, assessing the content validity and relevance of the items. They also further assess and review the applicability of the classifications attributed to each item. This resulted in 43 coarse items after synthesizing all atmospheric cues reported by respondents. These items have been translated into English and are presented in Table 2.

### 3.2 Purifying the measurement scale

**Refinement of instruments.** This coarse list is sent to the panel of experts to obtain content validity [79]. The panel conducts two workshops, reviewing each item one by one and assessing the applicability and representativeness of the measurement items toward the associated SFA. The panel decides to remove the entire dimension of "physical cues" for the following reasons: 1) Although the item "rapid increase in parcels" is indeed a physical cue, this item may have little effect on consumers' behaviors. 2) The items "logos and posters" and "colleagues' talking" are more suitable to be classified into the categories of social cues, although they present offline. 3) SDSFs is the online sales event; moreover, the physical cues only have three items. After removing and integrating the duplicative and inappropriate items, the initial items are then compiled based on the panel's constructive comments and, finally, reduced to 21 items belonging to 3 dimensions: 1) cultural cues (7 items), 2) social cues (7 items), and 3) online cues (7 items).

**Data collection.** In the third and fourth steps, the present study purifies the measures. The data collection is outsourced to a third-party company, wjx.cn (https://www.wjx.cn/), one of the largest online survey companies in China. There is such a large participant population involved in SDSFs that the technique of random sampling is needed. The company has the ability to do so because it has surveyed a great deal of respondents and volunteers over several years, including college students, white-collar workers, and urban residents.

**Table 2. Shopping festival atmospherics reported by respondents in the interviews.**

| Festival shopping atmospherics | Total counts |
|---|---|
| ***Cultural cues:*** | **54** |
| I want the ritual feeling. | 5 |
| Must empty the shopping cart at 12 a.m. that day. | 5 |
| I feel empty inside after 12 a.m. that day. | 3 |
| Held every year on the same day. | 6 |
| Prepare a long time for that day every year. | 9 |
| Shopping on Singles Day is a treatment for loneliness. | 3 |
| Buy! Buy! Buy! The consumerist culture is the point. | 7 |
| Must buy something! | 5 |
| Buy or feel sorry for myself. | 3 |
| I am pressed for time. I have to wait for the next year if I miss the deal. | 4 |
| Impulsive purchasing is tolerated that day. (social norm item) | 3 |
| Shopping during work time is tolerated that day. (social norm item) | 1 |
| ***Social cues:*** | **58** |
| Superstars share their experience of participating in the SDSFs. | 5 |
| Internet celebrities actively sell their own items. | 4 |
| I feel surrounded by SDSFs's popup ads, banner ads, embedded ads, etc. | 7 |
| Most friends in my WeChat circle are buying. | 10 |
| I will feel isolated if I do not join in shopping. | 6 |
| Lots of my connections in my WeChat circle are posting pictures of items. | 5 |
| "*Renao*" (happily crowded) environment. | 5 |
| Alibaba's shopping theme gala. | 3 |
| Interactive activities on *Xiaohongshu* (a popular e-commerce recommendation app). | 3 |
| Participants show their funny experiences on *Tik Tok* (a popular short video social app). | 3 |
| Buyer shows make me laugh. | 1 |
| Every year SDSFs is reported as the hot event. | 4 |
| So many shopping strategies are shared in my WeChat circle. | 5 |
| ***Physical cues:*** | **18** |
| I notice the rapid increase in parcels in my community. | 10 |
| I also notice many SDSFs logos and posters in supermarkets and stores. | 4 |
| I find my colleagues talk about SDSFs all day. | 4 |
| ***Online cues:*** | **56** |
| JD.com, NetEase, and most of China's e-commerce companies take part in the SDSFs. | 5 |
| The background color is replaced with red to highlight the joyous and festive atmosphere. | 7 |
| The SDSFs tag is clear to distinguish which items are on sale. | 5 |
| Every e-store replaces the portal with the SDSFs logo. | 3 |
| Special red envelopes and coupons for SDSFs are used. | 3 |
| Many online games increase participants' pleasure. | 1 |
| The e-stores are organized into several virtual divisions that are similar to physical promotional events. | 2 |
| So many items are on sale. | 5 |
| So many brands and vendors join in. | 5 |
| I hate price scams as some sellers raise the prices before. (negative cue) | 6 |
| I hate to be sold inferior items even if the price is indeed low. (negative cue) | 3 |
| Mobile purchasing is a good experience. | 3 |
| It is easy to learn how to buy items on sale. | 2 |
| I want to be treated fairly. | 3 |
| I want to have a festival that provides cheap and fine items. | 3 |

According to the survey instructions, the respondents decide whether to fill in the questionnaire voluntarily. They can scan the QR code or open the link to fill in the questionnaire through their mobile phones, and receive the reward after completing the questionnaire. The total sample size is 738. After deleting uncertain samples (e.g., incomplete answers, all the same answers, minors), the valid sample size is 686 (Six samples involving minors were deleted from the original data). Table 3 displays the profile of the respondents. As shown in Table 3., more than 98% of the respondents have participated in SDSFs, while approximately 80% participated in SDSFs almost every time. Approximately 75% of the respondents are from the young generation aged 20–40. However, there are still a quarter of the respondents who are middle-aged. In addition, more than half of the respondents have frequent chat experiences (5–19) during SDSFs. The demographics show that SDSFs is indeed an influential sales event attracting a large participant population.

**Table 3. Descriptive Statistics of Respondents (Sample 1 and Sample 2).**

| Variable | | Sample 1 (n = 343) | | Sample 2 (n = 343) | |
|---|---|---|---|---|---|
| Gender | Male | 157 | 45.77% | 166 | 48.40% |
| | Female | 185 | 53.94% | 177 | 51.60% |
| Age | 19–25 | 80 | 23.32% | 86 | 25.07% |
| | 26–30 | 73 | 21.28% | 75 | 21.87% |
| | 31–40 | 110 | 32.07% | 98 | 28.57% |
| | 41–50 | 65 | 18.95% | 71 | 20.70% |
| | 51–60 | 14 | 4.08% | 12 | 3.50% |
| | Greater than 61 | 1 | 0.29% | 1 | 0.29% |
| Education level | Middle school | 24 | 7.00% | 20 | 5.83% |
| | High school | 47 | 13.70% | 47 | 13.70% |
| | Bachelor's degree | 219 | 63.85% | 221 | 64.43% |
| | Graduate degree | 49 | 14.29% | 53 | 15.45% |
| | Post-graduate degree | 4 | 1.17% | 2 | 0.58% |
| Occupation | Working | 216 | 62.97% | 218 | 63.56% |
| | Student | 59 | 17.20% | 61 | 17.78% |
| | Freelancer | 42 | 12.24% | 36 | 10.50% |
| | Retired | 1 | 0.29% | 1 | 0.29% |
| | Other | 25 | 7.29% | 27 | 7.87% |
| Monthly income | Less than CNY 1000 | 31 | 9.04% | 41 | 11.95% |
| | CNY 1001–3000 | 39 | 11.37% | 37 | 10.79% |
| | CNY 3001–5000 | 55 | 16.03% | 47 | 13.70% |
| | CNY 5001–8000 | 115 | 33.53% | 111 | 32.36% |
| | CNY 8001–15000 | 66 | 19.24% | 76 | 22.16% |
| | CNY 15001–20000 | 31 | 9.04% | 23 | 6.71% |
| | Greater than CNY 20001 | 6 | 1.75% | 8 | 2.33% |
| Participation counts | Every time | 168 | 48.98% | 142 | 41.40% |
| | Almost every time | 113 | 32.94% | 141 | 41.11% |
| | Several times recently | 58 | 16.91% | 54 | 15.74% |
| | Never | 4 | 1.17% | 6 | 1.75% |
| Chat counts during SDSFs | Greater than 20 | 22 | 6.41% | 20 | 5.83% |
| | .10–19 | 59 | 17.20% | 55 | 16.03% |
| | 5–9 | 139 | 40.52% | 135 | 39.36% |
| | 1–4 | 116 | 33.82% | 126 | 36.73% |
| | Never | 7 | 2.04% | 9 | 2.62% |

In order to ensure the scientific and normative nature of the research, the collected data can be randomly divided into two parts. According to the research method and process of Dimitrova, Ilieva, and Stanev (2022), EFA and CFA were performed on the two parts of data respectively [80]. The proportions of gender, age, income, education, occupation, participation counts, and chat counts were maintained in the subsample. The demographic characteristics of the participants in the two randomly grouped sub-samples meet the requirements of the independence chi-square test. The numerical characteristics (mean and standard deviation) of the survey sample are shown in Table 4.

**Purifying the measurement.** The present study employs exploratory factor analysis (EFA) with a varimax rotation to determine the dimensions of the scales in sample1(n = 343). Items with factor loadings lower than 0.4 or cross-loaded on more than one factor are eliminated to ensure each attribute loads only on one factor [81]. In addition, the internal reliability of each factor is measured using Cronbach's alpha. Churchill (1979) suggests that a low alpha coefficient indicates the item has a low impact on the measurement of the construct [78]. Therefore, factors with Cronbach's alphas of less than 0.7 are eliminated [82].

The EFA suggests that the entire 21 items load on 3 factors just as classified above: social cues, cultural cues, and online cues. The first factor consists of 6 items with a Cronbach's alpha of 0.920. However, the item "Alibaba's shopping theme gala" is deleted from analysis due to its cross-loading on another factor. The second factor consists of 6 items with a Cronbach's alpha of 0.903. However, the item "Impulsive purchasing is tolerated that day" is deleted from analysis due to its cross-loading on another factor. In addition, the third factor consists of 7 items with a Cronbach's alpha of 0.899. By using Statistical Package for Social Sciences 21.0, the appropriateness of factor analysis is verified. The Kaiser-Meyer-Olkin (KMO) value is 0.902, which is expressed as "marvelous" (statistic>0.9) by Kaiser (1974) [83]. Bartlett's test of sphericity yields 4452.192 (p < 0.001), indicating the factor analysis is proper. The remaining 19 items are translated into English as listed in Table 5. The items include factor 1 (social cues, 6 items), factor 2 (cultural cues, 6

**Table 4. The measure of the sample data (n = 686).**

| Factor/Item | Sample 1 (n = 343) | | Sample 2 (n = 343) | |
|---|---|---|---|---|
| | Mean | Standard Deviations | Mean | Standard Deviations |
| Superstars and celebrities are active that day. | 5.379 | 1.470 | 5.466 | 1.416 |
| Most friends in my WeChat circle participate. | 5.207 | 1.469 | 5.440 | 1.406 |
| I feel the "Renao" (happily crowded) atmosphere. | 5.122 | 1.574 | 5.239 | 1.457 |
| Every year the SDSFs is reported as the hot event. | 5.172 | 1.542 | 5.309 | 1.508 |
| I receive many shopping strategies from Xiaohongshu, WeChat, and other social media. | 5.052 | 1.594 | 5.233 | 1.531 |
| Participants share their funny experiences. | 5.050 | 1.597 | 5.265 | 1.466 |
| I feel the SDSFs is a special day. | 4.606 | 1.754 | 4.633 | 1.832 |
| I have the ritual feeling, so I must buy something. | 4.962 | 1.648 | 4.921 | 1.612 |
| I prepare a long time for the day every year. | 4.799 | 1.718 | 4.831 | 1.790 |
| Shopping on Singles Day is a treatment for loneliness. | 5.012 | 1.656 | 5.149 | 1.790 |
| Buy! Buy! Buy! The consumerist culture is the point. | 5.096 | 1.649 | 5.058 | 1.657 |
| I am pressed for time. I have to wait for the next year if I miss the deal. | 5.073 | 1.589 | 5.082 | 1.750 |
| Most of China's e-commerce companies take part in the SDSFs. | 5.452 | 1.434 | 5.536 | 1.468 |
| The red background color and red envelopes highlight the joyous and festive atmosphere. | 5.531 | 1.456 | 5.504 | 1.535 |
| The logo, discount tag, and virtual division settings make the SDSFs more informative. | 5.052 | 1.661 | 5.105 | 1.602 |
| A large number of brands, vendors, and items take part in the SDSFs. | 5.003 | 1.563 | 5.047 | 1.616 |
| Many online games increase participants' pleasure. | 5.216 | 1.593 | 5.210 | 1.570 |
| The layout and process designed for both desktop and mobile shopping make the SDSFs more effective. | 5.239 | 1.599 | 5.359 | 1.575 |
| The sales are fair. | 5.044 | 1.575 | 5.090 | 1.591 |

items), and factor 3 (online cues, 7 items). The Cronbach's alpha coefficients for individual SFA factors range from 0.899 to 0.920, indicating highly reliable results.

### 3.3 Reanalysis of measures

*Confirmatory factor analysis.* Confirmatory factor analysis (CFA) was performed on sample 2 (n = 343) in AMOS21.0 to validate the factor structure extracted from the previous EFA. The degrees of freedom ratio (i.e., $\chi^2/df$) is 3.921 [84], comparative fit index (CFI) is 0.905, and non-normed fit index (NNFI) is 0.910 [85]. The root mean square error of approximation (RMSEA) is 0.092. However, the established criteria are CFI > 0.90, NNFI > 0.95, and RMSEA < 0.08 [84,86]. The CFA results do not meet the criteria very well. However, after deleting the item "Most of China's e-commerce companies take part in the SDSFs" (online cues), the results improve. It can be understood that this item is highly correlated with the item "a large number of brands, vendors, and items take part in the SDSFs". Ultimately, a total of 3 dimensions and 18 attributes remains in SFA.

*Criterion-related validity and cross-validation.* Table 5 shows the CFA results of the sample 2 (n = 343). After purification once more by the sample 1 (n = 343), the measurement model of the validation sample is found to be statistically significant. The results indicate a good model fit to the data, $\chi^2/df$=2.927, p < 0.001, CFI = 0.939, TLI = 0.926, and RMSEA = 0.075. In addition, as shown in Table 6., construct and discriminant validity concerns are not found (i.e., composite reliabilities > 0.7, average variance extracted (AVE) > 0.5) [87]. The square root of AVE (the bold diagonal number in the table) is higher than the correlation coefficient between structures, indicating that discriminant validity is satisfied [88], as shown in Table 7.

**Table 5. Exploratory Factor Analysis Results: Sample 1 (n = 343).**

| Factor/Item | Factor Loading | Cronbach's α |
|---|---|---|
| *Factor 1: Social cues* | | 0.920 |
| Superstars and celebrities are active that day. | 0.759 | |
| Most friends in my WeChat circle participate. | 0.854 | |
| I feel the "Renao" (happily crowded) atmosphere. | 0.798 | |
| Every year the SDSFs is reported as the hot event. | 0.854 | |
| I receive many shopping strategies from Xiaohongshu, WeChat, and other social media. | 0.847 | |
| Participants share their funny experiences. | 0.839 | |
| *Factor 2: Cultural cues:* | | 0.903 |
| I feel the SDSFs is a special day. | 0.747 | |
| I have the ritual feeling, so I must buy something. | 0.703 | |
| I prepare a long time for the day every year. | 0.773 | |
| Shopping on Singles Day is a treatment for loneliness. | 0.865 | |
| Buy! Buy! Buy! The consumerist culture is the point. | 0.784 | |
| I am pressed for time. I have to wait for the next year if I miss the deal. | 0.763 | |
| *Factor 3: Online cues:* | | 0.899 |
| Most of China's e-commerce companies take part in the SDSFs. | 0.717 | |
| The red background color and red envelopes highlight the joyous and festive atmosphere. | 0.727 | |
| The logo, discount tag, and virtual division settings make the SDSFs more informative. | 0.800 | |
| A large number of brands, vendors, and items take part in the SDSFs. | 0.800 | |
| Many online games increase participants' pleasure. | 0.781 | |
| The layout and process designed for both desktop and mobile shopping make the SDSFs more effective. | 0.756 | |
| The sales are fair. | 0.710 | |

## 4. Discussion and conclusions

The purpose of this study is to understand the atmospherics of a typical online shopping festival – China's Singles Day Shopping Festival. Many e-commerce researchers probably regard online sales events, such as Cyber Monday, Click Frenzy, and SDSFs, as large-scale promotions. Deep discounts and monetary incentives are the key features [9,15,18,35], which is consistent with the traditional theory of marketing mix. What is different is simply the vast scale. However, some extant studies note the scale itself leads to the special nature of these types of sales events. Herd effects [8,24] and hedonic shopping [1,14], which are not economic motivations, play an important role.

It seems that only Chinese e-commerce companies like to name online sales events festivals. The SDSFs is the most typical among them. No matter the festival-featured name, the SDSFs, initiated by Alibaba has continuously created a festival atmosphere every year. More and more Chinese researchers reported the effect of festival atmospherics

**Table 6. Performance of the measurement scale from CFA (sample 2, n = 343).**

| Items | Composite Reliability | AVE | Factor Loading | Critical Ratio | p |
|---|---|---|---|---|---|
| *Social cues:* | 0.926 | 0.678 | | | |
| Participants share their funny experiences. | | | 0.838 | — | *** |
| Superstars and celebrities are active that day. | | | 0.77 | 16.421 | *** |
| Most friends in my WeChat circle participate. | | | 0.835 | 18.877 | *** |
| I feel the "*Renao*" (happily crowded) atmosphere. | | | 0.846 | 19.35 | *** |
| Every year the SDSFs is reported as the hot event. | | | 0.852 | 19.525 | *** |
| I receive many shopping strategies from *Xiaohongshu*, *WeChat*, and other social media. | | | 0.795 | 17.66 | *** |
| *Cultural cues:* | 0.906 | 0.616 | | | |
| Buy! Buy! Buy! The consumerist culture is the point. | | | 0.83 | — | *** |
| I feel the SDSFs is a special day. | | | 0.711 | 14.542 | *** |
| I have the ritual feeling, so I must buy something. | | | 0.739 | 15.268 | *** |
| I prepare a long time for the day every year. | | | 0.813 | 17.655 | *** |
| Shopping on Singles Day is a treatment for loneliness. | | | 0.835 | 18.166 | *** |
| I am pressed for time. I have to wait for the next year if I miss the deal. | | | 0.772 | 16.177 | *** |
| *Online cues:* | 0.902 | 0.607 | | | |
| A large number of brands, vendors, and items take part in the SDSFs. | | | 0.81 | — | *** |
| The red background color and red envelopes highlight the joyous and festive atmosphere. | | | 0.688 | 13.488 | *** |
| The logo, discount tag, and virtual division settings make the SDSFs more informative. | | | 0.824 | 17.365 | *** |
| Many online games increase participants' pleasure. | | | 0.826 | 16.978 | *** |
| The layout and process designed both for desktop and mobile shopping make the SDSFs more effective. | | | 0.764 | 15.426 | *** |
| The sales are fair. | | | 0.752 | 15.102 | *** |

*** p < 0.001

**Table 7. Correlations of the measurement scale from CFA (sample 2, n = 343).**

| | Correlations | | |
|---|---|---|---|
| | Social cues | Cultural cues | Online cues |
| Social cues | **0.823** | | |
| Cultural cues | 0.490 | **0.785** | |
| Online cues | 0.411 | 0.585 | **0.779** |

on participants' shopping behaviors [1,2]. The shopping environment has proved to have effects on consumers' affect responses and behavioral outcomes since Kolter (1973) initiated the research stream [19]. There are many atmospherics-related studies in the literature: physical atmospherics in the retail context [40,47] and web atmospherics in the e-commerce context [56,59]. Cultural atmospherics are also taken into account in the research on festival shopping or holiday shopping [21,70].

The present study is the first attempt to integrate and capture the complete dimensions of atmospherics for an online sales event. Following Churchill (1979) eight steps to develop procedures, both qualitative and quantitative approaches are employed to develop a measurement scale, named "shopping festival atmospherics" [78]. The interview outline is extracted through a comprehensive review of the literature of shopping atmospherics. Then, a total of 23 interviews are conducted to explore the actual cues and participants' perceptions of SFA. Consistent with the literature, the respondents provide information classified into 4 categories: cultural cues, social cues, physical cues, and online cues. These cues give us a more comprehensive idea of the atmospherics of the SDSFs than has ever before been perceived. Following the rigorous development procedures, the initially coarse items are refined and purified step by step until finally, the measurement scale of SFA is formed with eighteen attributes divided into three factors.

## 4.1 Theoretical implications

This research has a number of theoretical implications. First, according to the S-O-R theory, which focuses on "environmental stimuli can trigger individual emotional responses", combined with previous research findings, a shopping festival atmosphere scale was developed. In recent years, with the widespread attention of scholars on online shopping atmosphere, scholars have described the connotation of shopping festival atmosphere from different perspectives [24,26]. The research findings of this paper not only establish the concept of shopping festival atmosphere, but also expand its theoretical connotation and application.

Secondly, the SFA scale developed in this paper is different from previous measures of atmosphere because it contains three components: cultural, social, and online atmosphere, and it is described in detail. The shopping festival atmosphere not only adds the concept of "festival" to the original online shopping atmosphere, but also takes into account China's unique cultural background and the most extensive online and offline social interaction. This concept extends the application of atmosphere research to the phenomenon of "online shopping festival". The connotation of the shopping festival atmosphere identified in this study is discussed as follows:

Cultural clues refer to the particularity and sense of ritual that must be consumed in online shopping festivals. SDSFs was initiated from Singles Day, an unofficial festival. However, the festival creates a custom for the SDSFs. It is held on November 11 every year. This is a strong cultural hint in many participants' eyes. They treat SDSFs as a special day. They have the custom or habit of accomplishing what must be done. Just as Zhao and Wan (2017) found in their study that participating in SDSFs is a ritual event [35]. In addition, they need to prepare in advance for these "must-do" behaviors. On the other hand, the consumerist culture is booming, associated with the SDSFs in China. The Confucian values of "keeping face" and "listening to others" exert a moderating influence on the effect of the atmosphere of the online shopping festival on purchase intention [24]. The happily crowded environment ("Renao" in the Chinese language) is a positive atmospheric cue for Chinese participants. This type of cultural cue gives participants a symbolic link between the SDSFs and the "Buy! Buy! Buy!" slogan.

Social cues mean that participants actively participate in the most extensive topic communication and knowledge sharing during the SDSFs, both offline and online. The extensive social interaction on the shopping day (such as topic discussion, shopping communication, experience sharing, etc.) has gradually become the biggest feature of Singles Day shopping day [9,25]. Through online and offline advertising, single consumers can establish a sense of identity with Singles Day [35]. Furthermore, Chen and Li (2020) demonstrated that the festive atmosphere exerts an influence on consumers, which is shaped by the social medias and the people around them [9]. All of these social environments should

ultimately result in a crowded atmosphere that is safe, secure and happy. Otherwise, participants may experience negative emotions and misbehave [15,18].

Online cues are the evaluation of the comprehensive service of the network platform by the consumers participating in the online shopping festival. As an online sales event, the SDSFs had to make changes for special days. This study found that in addition to the traditional online climate of information, efficiency and entertainment, the fair climate was also a new feature perceived by participants. They need very good deals [25]. They don't want to be cheated by artificially inflated prices and they don't want to go to great lengths to play the coupon game. In addition, the authors found that size itself is an attribute of online atmosphere. Studies by Tzeng, Ert, Jo et al. (2020) show that product information and product quality are prerequisites for consumers to feel satisfied after participating in online shopping festivals [10]. The website informativeness and effectiveness of information content can help to improve consumer satisfaction and purchase intention [38]. Participants' perception of the promotional scale was that they expected more opportunities to get what they really wanted at a lower price than usual, more opportunities to avoid restrictions on the discounted items they wanted, and ultimately more benefits from SDSFs.

## 4.2 Managerial implications

This study has established a reliable scale for measuring the "shopping festival atmosphere", and has also provided managers with a tool for objectively understanding consumers' perception of the shopping festival atmosphere. The results of this study have significant implications for the practice of management, which can be grouped into many areas: Firstly, managers are able to utilize this scale to ascertain whether the shopping festival atmosphere has been successfully created, particularly when the SFA is low. Subsequently, managers are able to modify their strategies in accordance with the aforementioned findings. Secondly, the utilization of cultural, social and online cues can also serve as a valuable means of evidence for managers seeking to establish a shopping festival atmosphere. Thirdly, these cues can be integrated with actual circumstances to develop innovative designs that enhance consumers' purchase intention during the shopping festival. Ultimately, the findings of this study offer valuable insights that can inform the successful implementation of online shopping festivals in other countries, particularly in light of the growing prevalence of such events in China and the global convergence of e-commerce platforms.

## 5. Limitations and future research

In summary, this study developed a "shopping festival atmosphere" measurement scale and extracted some novel atmosphere cues from the SDSFs. Given the dearth of research in this field, however, some limitations were also identified that need to be addressed and considered.

Firstly, the questionnaire data in this paper was collected only once. However, recent studies on scale development have recommended that questionnaires be collected at least twice. According to the reasonable practice of scale development in previous literature, this paper randomly divided the collected questionnaires into two parts, and carried out EFA and CFA, respectively to ensure the standardization of scale development.

Secondly, the questionnaire in this study was collected in the form of online response, and a small amount of questionnaire data would become invalid due to random filling by the respondents. At the same time, the questionnaire samples collected in this study were only Chinese, and no foreigners were involved.

Thirdly, regarding the scale development of shopping festival atmosphere, this study focused on the efforts of shopping platforms, consumer behavior and feelings. However, this study ignores the influence of different attributes of goods (such as hedonic and practical) on the atmosphere of the shopping festival, as well as the influence on shopping psychology and behavior.

Some scholars have studied "shopping festival atmosphere" (SFA) and found that this factor has a negative impact on consumption intention and behavior. The scale development of SFA in this study provides a theoretical basis for the

future research on the consumption behavior of this factor. In the future, the research on SFA can draw on the scale of this study, and analyze the connotation of online or offline SFA. The online shopping festival has entered the era of live e-commerce, and the "shopping festival atmosphere" under this situation has a new connotation, which needs to be further explored. Many scholars have studied the shopping atmosphere of live e-commerce, which also provides a basis for future research.

## Supporting information

**S1 Data. Questionnaire Data (Survey on SDFs).**
(XLSX)

## Acknowledgments

The authors would like to thank Professor Chunling Shao for her contributions in revising and supervising the article. The authors are also grateful to the anonymous reviewers for their constructive feedback and to all survey respondents for their participation.

## Author contributions

**Investigation:** Jingyu Li.

**Methodology:** Qiuyang Gu.

**Resources:** Xiaoming Wang.

**Supervision:** Xiaoming Wang.

**Validation:** Qiuyang Gu.

**Writing – review & editing:** Jingyu Li.

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
