## [Decision Letter · Decision Letter 0]

25 June 2024

PONE-D-24-00459Shopping festival atmospherics of China’s Singles Day Shopping Festival and participants’ perception: Scale development and validationPLOS ONE

Dear Dr. Li,

Thank you for submitting your manuscript to PLOS ONE. After careful consideration, we feel that it has merit but does not fully meet PLOS ONE’s publication criteria as it currently stands. Therefore, we invite you to submit a revised version of the manuscript that addresses the points raised during the review process.

We look forward to receiving your revised manuscript.

Kind regards,

Abaid Ullah Zafar, Ph.D.

Academic Editor

PLOS ONE

Journal Requirements:

2. You indicated that ethical approval was not necessary for your study. We understand that the framework for ethical oversight requirements for studies of this type may differ depending on the setting and we would appreciate some further clarification regarding your research. Could you please provide further details on why your study is exempt from the need for approval and confirmation from your institutional review board or research ethics committee (e.g., in the form of a letter or email correspondence) that ethics review was not necessary for this study? Please include a copy of the correspondence as an ""Other"" file.

"The Philosophy and Social Science Planning Project of Zhejiang Province, No. 23NDJC414YBM"

5. We note that your Data Availability Statement is currently as follows: All relevant data are within the manuscript and its Supporting Information files.

**Additional Editor Comments:**

I wanted to inform you that we have completed the review of your article. After careful consideration, we have reached a major revision decision regarding its publication.

We appreciate the effort and time you invested in this submission. Attached, you will find the detailed feedback from our reviewers that guided our decision. Please note that this revision will be risky, as the article focuses on scale development and the reviewers have raised significant concerns. You are advised to carefully address all the concerns.

Thank you for considering our journal for your work. We look forward to your revised submission.

Reviewers' comments:

Reviewer's Responses to Questions

**Comments to the Author**

1. Is the manuscript technically sound, and do the data support the conclusions?

Reviewer #1: Partly

Reviewer #2: Yes

2. Has the statistical analysis been performed appropriately and rigorously? 

Reviewer #1: Yes

Reviewer #2: Yes

3. Have the authors made all data underlying the findings in their manuscript fully available?

Reviewer #1: Yes

Reviewer #2: Yes

4. Is the manuscript presented in an intelligible fashion and written in standard English?

Reviewer #1: Yes

Reviewer #2: Yes

5. Review Comments to the Author

Reviewer #1: The paper discusses atmosphere scale development based on the successful online shopping SDSF event. The topic is relevant and the methodological approach is basically adequate. However I have some major concerns about the presentation in the paper and the underlying data collection and analysis:

1. The literature relies mainly on elder articles. Many current and relevant publications are not reflected. So, e.g., Cheng and Chen (2023) discuss an atmosphere scale development in a detailed and methodologically advanced manner. Also, the author cites many times DeVellis (2003) where as DeVellis and Thorpe (2021), the advanced and extended fifth edition of this standard book privdes advanced insights. Also "Kolter 1973" is cited, but the author wants to cite Kotler 1973, the famous atmosphere article by the famous marketing professor (:O). Few scientific articles cited in the paper are from 2015 onwards (exactly: 10), the paper needs a complete refresh of its references and consequently a discussion of the state of the art based on this refreshing.

2. The author refers to Churchill and others for scale development. However, only one survey is conducted in contradiction to Chruchill's two surveys (after eliminating and extending the item batteries). I miss a discussion why this is adequate. Just to mention Cheng and Chen (2023): They use two surveys, the second one is based on a modified questionnaire. Also, the search for items in the construct development phase during the semi-structured interviews is not convincing: The interviewers focus on the SDSF event concerning atmosphere (positive aspects: "Can you list some

atmospheric cues you feel during the SDSF?"). They completely miss the aspects that participants miss during an SDSF event. Again, look at Cheng and Chen (2023) for improvements: They ask what "defines" "good" atmosphere at an event.

3. The author refers to standard models in marketing (SOR model) but I miss a clear reference to the origins of these models (e.g., Woodworth, or Howard and Sheth).

4. The paper discusses scale developement for atmosphere at the SDSF event. However, for me, it is unclear, whether this scale could also be used to measure atmosphere at other online and offline events. Usually, if you develop a standard scale, it is used to compare competitors, users, and the like. The author should more clearly explain why it is so important to develop a scale that is focused on one specific event.

References

Cheng, T. M., & Chen, M. T. (2023). Creative atmosphere in creative tourism destinations: Conceptualizing and scale development. Journal of Hospitality & Tourism Research, 47(3), 590-615.

DeVellis, R. F., & Thorpe, C. T. (2021). Scale development: Theory and applications. Sage publications, Fifth Edition.

Reviewer #2: Thank you for conducting such interesting research. Although it has several advantages, there are also some issues that you should resolve before further processing.

Introduction::

• The introduction provides a good overview of the importance and growth of China's Singles Day Shopping Festivals (SDSFs), highlighting their massive scale and increasing popularity.

• It identifies the lack of systematic analysis of the atmospheric elements of these major online sales events as a research gap.

• The objectives of the study, which are to develop a measurement scale for "shopping festival atmospherics" (SFA) and examine participants' perceptions of the atmospherics, are clearly stated.

But there are some Weaknesses:

• The introduction could be strengthened by providing more context on the existing research on shopping atmospherics, both in offline and online settings. This would help situate the current study within the broader literature. I suggest adding the following research to your paper to improve this section:

Shafiee, M. M., & Es-Haghi, S. M. S. (2017). Mall image, shopping well-being and mall loyalty. International Journal of Retail & Distribution Management, 45(10), 1114-1134.

Shafiee, M. M., & Bazargan, N. A. (2018). Behavioral customer loyalty in online shopping: The role of e-service quality and e-recovery. Journal of Theoretical and Applied Electronic Commerce Research, 13(1), 26-38.

Tabaeeian, R. A., & Mohammad Shafiee, M. (2023). Identifying factors affecting the motivation of games users in social networks and their impact on the user attitude and shopping intention. New Marketing Research Journal, 12(4), 51-68.

• The rationale for focusing specifically on SDSFs, rather than online shopping festivals in general, could be elaborated on further.

• Expand the literature review in the introduction to give a more comprehensive overview of the existing research on shopping atmospherics and how it has been applied in online and festival settings.

• Clearly articulate the theoretical and practical significance of developing a scale to measure SFA.

• Provide a more detailed justification for the choice of SDSFs as the context for this study.

Literature Review::

• The literature review covers the key concepts and theories relevant to the study, including the definition of atmospherics and its application in retail and festival settings.

• The review discusses the existing research on the success factors and motivations of consumers for participating in online shopping festivals, providing useful background information.

• However, the literature review could be more comprehensive, incorporating a wider range of studies on shopping atmospherics, both in offline and online contexts. Please use previous mentioned sources to improve this section.

• The review could also delve deeper into the specific atmospheric cues that have been identified in the literature for offline and online shopping environments, as well as festival settings.

• Expand the literature review to include a more thorough examination of the broader research on shopping atmospherics, highlighting the key dimensions and cues that have been identified in previous studies.

• Discuss how the atmospherics of online shopping environments and festival settings may differ from traditional offline retail settings, and the implications for the current study.

• Identify any gaps or limitations in the existing literature that the current study aims to address.

I also suggest using and adding the following sources to update your sources in this section:

Shafiee M. M., Yavari, Z. & Ghorbanian, P. (2015). Ranking of Selected Convenience Stores of Isfahan Based on Store Image Dimensions with Group AHP Technique. Journal of Operational Research and Its Applications, 12(46), 35-47.

Ghorbanian, P., Yavari, Z., & Mohammad Shafiee, M. (2015). Analysis of retailer equity based on selected store image dimensions (Case study: Refah, CityCenter (HyperStar) & Kowsar stores). New Marketing Research Journal, 5(3), 143-160.

Mohammad Shafiee, M., & Ahghar Bazargan, N. (2016). Electronic trust of customers to online stores with a risk reduction approach. Journal of Karafan, 6(10), 113-122.

Mohammad Shafiee, M., Bazargan, N. A., & Kazeminia, A. (2017). Modeling Customer Electronic Trust in Online Stores: A Risk Reduction Approach. 11th International Conference on e-Commerce in Developing Countries (ECDC), April, Isfahan, Iran.

Nurani, M., Rezaei Dolatabadi, H., & Mohammad Shafiee, M. (2021). Designing a store brand competitiveness model based on environmental stimuli In chain stores. Journal of Business Management Perspective, 19(44), 13-40.

Nurani Kutenaee, M., Rezaei Dolatabadi, H., & Mohammad Shafiee, M. (2021). Modeling the Competitiveness of a Store Brand based on the pattern of Environmental Estimuli in Chain Stores: A mixed approach. Management Research in Iran, 25(2), 151-182.

Methodology:

• The study employs a rigorous, multi-step approach to scale development, including both qualitative and quantitative methods.

• The use of multiple data sources (literature review, focus groups, and online survey) to generate and refine the scale items is a strength.

• The validation of the scale through confirmatory factor analysis is appropriate and well-executed.

• But the methodology section could provide more details on the sample characteristics and data collection procedures for the quantitative phase.

• The rationale for the specific statistical techniques used (e.g., exploratory factor analysis, confirmatory factor analysis) could be elaborated on further.

• Expand the methodology section to include more details on the sample composition, data collection methods, and data analysis procedures used in the quantitative phase of the study.

• Provide a clearer explanation of the decision-making process behind the selection of the statistical techniques employed.

• Consider including information on the reliability and validity assessments conducted for the final SFA scale.

Results and Discussion:

• The results section presents the key findings from the scale development and validation process in a clear and concise manner.

• The discussion section effectively interprets the findings, highlighting the theoretical and practical implications of the study.

• The identification of novel atmospheric cues, such as the "happily crowded environment" and "fair shopping environment", is a valuable contribution.

Weaknesses:

• The discussion could be strengthened by more explicitly connecting the findings to the existing literature on shopping atmospherics and festival settings.

• The limitations of the study and directions for future research could be expanded upon.

• Enhance the discussion by drawing stronger links between the identified SFA dimensions and cues and the broader literature on shopping atmospherics.

• Discuss the potential cultural and contextual factors that may have influenced the emergence of the novel atmospheric cues.

• Expand the limitations section to acknowledge the study's boundaries and provide more detailed recommendations for future research.

Overall, the paper presents a rigorous and comprehensive approach to developing a measurement scale for the atmospheric elements of China's Singles Day Shopping Festivals. The strengths of the study include the multi-method scale development process, the identification of novel atmospheric cues, and the clear discussion of the theoretical and practical implications. To further improve the paper, the authors could strengthen the literature review, provide more details on the methodology, and enhance the discussion by more explicitly connecting the findings to the existing research.

6. PLOS authors have the option to publish the peer review history of their article (what does this mean? ). If published, this will include your full peer review and any attached files.

**Do you want your identity to be public for this peer review?** For information about this choice, including consent withdrawal, please see our Privacy Policy .

Reviewer #1: No

Reviewer #2: **Yes: ** Majid Mohammad Shafiee

---

## [Author Response · Author response to Decision Letter 1]

28 Sep 2024

Response to Editor

Dear Editor,

Thank you very much for giving us an opportunity to revise our manuscript. We appreciate the editor and reviewers sincerely for their constructive comments and suggestions to our manuscript entitled “Shopping festival atmospherics of China’s Singles Day Shopping Festival and participants’ perception: Scale development and validation” (PONE-D-24-00459)

Those comments are all· valuable and very helpful for revising and improving our paper, as well· as the important guiding significance to our following research. We have presented detailed response to reviewers’ comments. According to the reviewers’ comments, we have made a careful revision on the original manuscript. All revise contents are marked in yellow in the revised manuscript.

The comments and responses are listed in next pages.

Journal Requirements:

Question 1

Response 1: Thank you very much for your kind reminder and the paper submission template.

We have carefully revised the paper to meet PLOS ONE's style requirements.

Question 2

You indicated that ethical approval was not necessary for your study. We understand that the framework for ethical oversight requirements for studies of this type may differ depending on the setting and we would appreciate some further clarification regarding your research. Could you please provide further details on why your study is exempt from the need for approval and confirmation from your institutional review board or research ethics committee (e.g., in the form of a letter or email correspondence) that ethics review was not necessary for this study? Please include a copy of the correspondence as an ""Other"" file.

Response 2: Thank you for your kind reminder. We will explain it from the following aspects:

Firstly, this study attempts to extend the dimensions and attributes of ambiance in online promotions. Secondly, the sample data of the study is only about the consumption behavior and intention of the online shopping festival, and does not involve the data of the medical field.

Finally, in order to meet the ethical norms of the journal and ensure the scientific nature of the research, the Scientific Research Office of "Yiwu Industrial and Commercial College" issued an "Ethics Committee review approval". I hope I can get your approval. See Annex 1 for details.

Question 3

Please provide additional details regarding participant consent. In the ethics statement in the Methods and online submission information, please ensure that you have specified (1) whether consent was informed and (2) what type you obtained (for instance, written or verbal, and if verbal, how it was documented and witnessed). If your study included minors, state whether you obtained consent from parents or guardians. If the need for consent was waived by the ethics committee, please include this information.

Response 3: Thank you very much for your attention and reminder of the research sample. We have screened the questionnaire data one by one and made the following adjustments.

(1) First of all, the survey is voluntary. Secondly, before filling in the questionnaire, the respondents were given instructions to fill in the questionnaire, including the purpose and main content of the questionnaire. The respondents could decide whether to participate in the social survey or not. Finally, the privacy protection of participants is explained.

(2) We designed the questionnaire and entrusted it to the largest questionnaire service company in China—wjx.cn (https://www.wjx.cn/) for questionnaire collection. Details are as follows:

1) The authors designed the questionnaire according to the purpose of the study.

2) Using the service function of the wjx.cn (https://www.wjx.cn/), the questionnaire can be directly read and filled out on the mobile phone.

3�Entrust wjx.cn (https://www.wjx.cn/) (the largest questionnaire service platform in China) to sign the service agreement to determine the demographic attributes of the questionnaire collection, the number of questionnaires and the reward of the questionnaire.

4�According to the reading of the survey instructions, the respondents decide whether to fill in the questionnaire voluntarily. They can scan the QR code or open the link to fill in the questionnaire through their mobile phones, and receive the reward after completing the questionnaire.

5�Finally, the collected questionnaires were preprocessed, and the unreasonable questionnaires such as incomplete questionnaires and extreme answers were eliminated. The final valid questionnaires were used for data statistical analysis.

(3) According to the feedback of the wjx.cn (https://www.wjx.cn/), after answering each valid questionnaire, the respondents will receive a certain red envelope reward. At the same time, the IP address and filling time of each questionnaire can be traced to the source, so as to ensure the authenticity and validity of the questionnaire.

(4) A total of 6 minors participated in this questionnaire survey. In order to avoid the ethical issues involved in the survey data of minors, after careful consideration, we decided to eliminate the data involving six minors and then conduct the data analysis again.

(5) Six questionnaires from minors were removed from the study sample data, leaving 686 valid questionnaires.

(6) The following are the instructions for filling out the questionnaire:

Dear respondents

Hello! In recent years, with the upgrading of online shopping consumption demand, the online shopping festival with the "Double 11 online shopping Carnival" has shown great consumption potential. As college teachers, we conducted a social survey on consumers' perception of "shopping festival atmosphere" in the "Double 11 Online Shopping Carnival". We hope to form an effective scale for "shopping holiday atmosphere" through statistical analysis of the collected questionnaire data. The data obtained from this questionnaire is only used for academic research. It will not disclose your personal privacy and personal views, and will not be used for any commercial purposes. Please feel free to fill in the questionnaire. Finally, in order to thank you for your support for this social survey, after filling out the questionnaire, there will be a red envelope reward.

Question 4

Thank you for stating the following financial disclosure:

"The Philosophy and Social Science Planning Project of Zhejiang Province, No. 23NDJC414YBM"

Response 4: Thank you very much for your kind reminder to us on this issue, for which we make the following explanation.

It should be noted that, on the one hand, the head of the funded project is Professor Shao Chunling. On the other hand, the first author of this paper is a core member of this funded project.

The funder, Professor Chunling Shao, did not play a role in the study design, data collection and analysis, decision to publish, or preparation of the manuscript. However, in the process of paper revision, Professor Shao Chunling supervised and funded the paper layout fee. Therefore, the authors of this paper unanimously agreed to add Professor Shao Chunling to the author sequence of this paper.

Question 5

We note that your Data Availability Statement is currently as follows: All relevant data are within the manuscript and its Supporting Information files.

If your submission does not contain these data, please either upload them as Supporting Information files or deposit them to a stable, public repository and provide us with the relevant URLs, DOIs, or accession numbers. For a list of recommended repositories, please see

https://journals.plos.org/plosone/s/recommended-repositories.

Response 5: Thank you very much for your suggestions on the data analysis part. We have also made the following adjustments.

Firstly, we will upload the data set used in this study as required.

Second, detailed data reports (such as mean and standard deviation) have been added to this study in section 3.2 (page 13) Table4.

Finally, our attempts to submit the data in the data repository were always unsuccessful, so we submitted a data link (url: https://pan.quark.cn/s/3dce718cd231 ), through which we could directly access the questionnaire data used in this study.

Question 6

Please include captions for your Supporting Information files at the end of your manuscript, and update any in-text citations to match accordingly. Please see our Supporting Information guidelines for more information: http://journals.plos.org/plosone/s/supporting-information.

Response 6: Thank you very much for your many suggestions during this submission process. We must carefully revise the paper according to the requirements of the editors and reviewers, and upload the repaired paper within the prescribed time.

Response to Reviewer 1 Comments

Thank you very much for taking the time to review this manuscript. These opinions help to improve the academic rigor of our article. Please find the detailed responses below and the corresponding revisions highlighted in the re-submitted files.

Reviewer #1: The paper discusses atmosphere scale development based on the successful online shopping SDSF event. The topic is relevant and the methodological approach is basically adequate. However, I have some major concerns about the presentation in the paper and the underlying data collection and analysis:

Question 1

The literature relies mainly on elder articles. Many current and relevant publications are not reflected. So, e.g., Cheng and Chen (2023) discuss an atmosphere scale development in a detailed and methodologically advanced manner. Also, the author cites many times DeVellis (2003) where as DeVellis and Thorpe (2021), the advanced and extended fifth edition of this standard book provides advanced insights. Also "Kolter 1973" is cited, but the author wants to cite Kotler 1973, the famous atmosphere article by the famous marketing professor (:O). Few scientific articles cited in the paper are from 2015 onwards (exactly: 10), the paper needs a complete refresh of its references and consequently a discussion of the state of the art based on this refreshing.

Response 1: I am very glad that you have provided me with two important references, which have provided great help for the revision of this paper.

Firstly, we carefully listen to your suggestions and update the literature cited in the paper in a timely manner by reading the relevant literature in the past 5 years.

Secondly, according to the relevant literature, theoretical basis and research methods of the research topic in this paper, we have made a lot of changes in the introduction, literature review and discussion of research results.

Finally, Thank you again for the paper (Cheng and Chen, 2023) and the latest reference book (DeVellis and Thorpe, 2021). The specific changes are as follows.

1�In the introduction, first of all, the previous content and paragraph were adjusted. Secondly, the research literature on "China's Singles Day Shopping Festival" and "festival atmosphere" in the past five years is focused on. Finally, a description of the value of the research is added.

Page1 line 40-50

Improving the variety of promotion categories and the interest of promotion activities are very effective promotion strategies during online shopping festival [9]. Information quality, product quality and cost savings can significantly improve consumer satisfaction with online shopping festivals [10]. Later, researchers found that the Double 11 online shopping festival was not only a promotional activity, but also an entertainment for participants. Collaboration between Chinese e-commerce platforms and merchants in terms of product promotion and atmospheric marketing strategies is key to the success of the SDSFs [9]. Xu, Wang, and Zhao (2020) posit that SDSFs creates unique shopping scenarios that can more effectively stimulate consumers' emotions and influence their shopping behaviors [11]. Hedonistic shopping has become the main motivation for consumers to participate in online shopping festivals, in addition to saving money [12].

Page2-3 line71-99

The influence of shopping festival atmospheres on consumer behavior in SDSFs has also been the subject of research by scholars in the field. In a study conducted by Yang, Li, and Zhang (2018), it was discovered that the atmosphere surrounding SDSFs was negatively correlated with consumers' intentions to engage in sustainable consumption [23]. This atmosphere has become SDSFs most important marketing tool for stimulating consumer purchases [24]. The shopping festival atmosphere plays a positive moderating role between promotional stimulation and satisfaction, as well as between satisfaction and the intention to continue participating [25]. Despite the considerable attention that research on the atmosphere of online shopping festivals has attracted, there is still no consensus on the exact connotations of this phenomenon. There are no clear conclusions about the difference between the atmosphere of online shopping festivals and that of traditional shopping environments, and there is a lack of scientifically valid scales to describe this unique atmosphere.

"SDSFs" created by Alibaba is the earliest online shopping festival in China (2009.11.11), with the highest popularity, the most extensive consumer participation and the most influential [8, 10]. The success of other online shopping festivals (e.g. JD 6.18) also draws lessons from the successful experience of SDSFs. The rapidly growing sales data of "SDSFs" over the years and the popular discussion on social platform [1, 23]. SDSFs has undergone a significant transformation, evolving from a focus on simple price promotions to the development of comprehensive marketing campaigns that prioritize the creation of an entertaining and interactive atmosphere [26]. Given the relatively standardized and homogeneous nature of e-commerce activities worldwide, the findings of the survey on Chinese e-commerce platforms can be extrapolated to a number of other platforms globally. This offers valuable insights and recommendation

---

## [Decision Letter · Decision Letter 1]

6 May 2025

Shopping festival atmospherics of China's Singles Day Shopping Festival and participants' perception: Scale development and validation

PONE-D-24-00459R1

Dear Dr. Li,

We’re pleased to inform you that your manuscript has been judged scientifically suitable for publication and will be formally accepted for publication once it meets all outstanding technical requirements.

Kind regards,

Academic Editor

PLOS ONE

Additional Editor Comments (optional):

Reviewers' comments:

Reviewer's Responses to Questions

**Comments to the Author**

1. If the authors have adequately addressed your comments raised in a previous round of review and you feel that this manuscript is now acceptable for publication, you may indicate that here to bypass the “Comments to the Author” section, enter your conflict of interest statement in the “Confidential to Editor” section, and submit your "Accept" recommendation.

Reviewer #1: All comments have been addressed

Reviewer #3: (No Response)

2. Is the manuscript technically sound, and do the data support the conclusions?

Reviewer #1: Yes

Reviewer #3: (No Response)

3. Has the statistical analysis been performed appropriately and rigorously? 

Reviewer #1: Yes

Reviewer #3: (No Response)

4. Have the authors made all data underlying the findings in their manuscript fully available?

Reviewer #1: Yes

Reviewer #3: (No Response)

5. Is the manuscript presented in an intelligible fashion and written in standard English?

Reviewer #1: Yes

Reviewer #3: (No Response)

6. Review Comments to the Author

Reviewer #1: Congratulations, the paper has improved. All the best for your current and future research, looking forward to additional papers from you.

Reviewer #3: Thank you for the opportunity to review this manuscript. After a thorough evaluation, I find the paper to be well-written, methodologically sound, and making a valuable contribution to the literature. The research objectives are clearly defined, the methodology is appropriate, and the findings are insightful and well-presented.

7. PLOS authors have the option to publish the peer review history of their article (what does this mean? ). If published, this will include your full peer review and any attached files.

**Do you want your identity to be public for this peer review?** For information about this choice, including consent withdrawal, please see our Privacy Policy .

Reviewer #1: No

Reviewer #3: No

---

## [Editor Report · Acceptance letter]

PONE-D-24-00459R1

PLOS ONE

Dear Dr. Li,

I'm pleased to inform you that your manuscript has been deemed suitable for publication in PLOS ONE. Congratulations! Your manuscript is now being handed over to our production team.

Kind regards,

on behalf of

Dr. Abaid Ullah Zafar

Academic Editor

PLOS ONE